# Total Neoadjuvant Therapy in Locally Advanced Rectal Cancer: Insights from the Western Australian Context

**DOI:** 10.3390/diseases12100257

**Published:** 2024-10-17

**Authors:** Oliver Oey, Chak Pan Lin, Muhammad Adnan Khattak, Thomas Ferguson, Mary Theophilus, Siaw Sze Tiong, Sayed Ali, Yasir Khan

**Affiliations:** 1UWA Medical School, University of Western Australia, Perth, WA 6009, Australia; oliver.oey@health.wa.gov.au; 2Department of Medical Oncology, Sir Charles Gairdner Hospital, Nedlands, WA 6009, Australia; 3Mater Hospital, Brisbane, QLD 4101, Australia; benchakpanlin@gmail.com; 4Department of Medical Oncology, Fiona Stanley Hospital, Murdoch, WA 6150, Australia; thomas.ferguson@health.wa.gov.au; 5Department of General Surgery, St John of God Midland Hospital, Midland, WA 6056, Australia; mary.theophilus@sjog.org.au; 6Icon Cancer Centre, Midland, WA 6056, Australia; eve.tiong@icon.team; 7Department of Medical Oncology, St John of God Midland Hospital, Midland, WA 6056, Australia; sayed.ali@health.wa.gov.au; 8Peel Health Campus, Mandurah, WA 6210, Australia

**Keywords:** total neoadjuvant therapy, TNT, locally advanced rectal cancer, LARC, cancer treatment, chemoradiotherapy, chemotherapy, radiotherapy

## Abstract

Background: Recent studies have associated total neoadjuvant therapy (TNT) with better treatment adherence, decreased toxicity, improved complete clinical response and anal sphincter preservation rates in patients with locally advanced rectal cancer (LARC). However, real-world experience with TNT in the management of LARC remains limited. Aim: This study aimed to evaluate the efficacy and safety outcomes of TNT for LARC in Western Australia. Methods: Patients with LARC (cT2-4 and/or cN1-2) who underwent induction chemotherapy followed by neoadjuvant chemoradiotherapy or neoadjuvant chemoradiotherapy followed by consolidation chemotherapy, followed by surgery were recruited from two hospitals in Western Australia. Efficacy outcomes assessed included clinical response (complete, partial, no response), and pathologic complete response (pCR) rate, R0 resection rate, and R1 resection rate were evaluated. Those patients who achieved clinical complete response following TNT were given the option of active surveillance. The safety and tolerability of TNT were assessed. Results: 32 patients with LARC were treated with TNT. In total, 17 patients (53%) received chemoradiotherapy followed by consolidation chemotherapy and 15 patients (47%) received induction chemotherapy followed by chemoradiotherapy. Nine (28%) of the patients with LARC treated with TNT had a complete clinical response, twenty-one (66%) patients had a partial clinical response, and two (6%) patients had no response to TNT. Of the 32 patients, 27 (84%) underwent surgery. There was a 100% R0 resection rate. The pCR rate was 15%. pCR, clinical response, and the R0 resection rate were similar between the two TNT regimens. TNT was well tolerated, with the majority of patients (88%) completing the chemotherapy course with grade 1 and 2 adverse effects. Conclusions: In conclusion, TNT emerges as a promising approach for the management of LARC. However, further research is warranted to refine the optimal TNT protocols, determine its long-term outcomes, and identify patient populations who would benefit the most from this innovative therapeutic strategy.

## 1. Introduction

Over the past decade, the established treatment protocol for locally advanced rectal cancer (LARC) has comprised neoadjuvant therapy (either radiation or chemoradiotherapy), followed by total mesorectal excision (TME) and adjuvant systemic chemotherapy thereafter [1]. Despite multiple efforts to potentiate preoperative chemoradiation regimens, pathological complete response (pCR) and distant control remains suboptimal, with rates of about 10–15% and 25–35%, respectively [1]. Poor adherence to adjuvant chemotherapy remains a primary contributor to recurrence, as only about half of patients successfully complete postoperative adjuvant chemotherapy, thereby amplifying the risk of micrometastasis [2].

In recent years, total neoadjuvant therapy (TNT) has shown promise in LARC patients. TNT represents an innovative strategy for LARC. The aim of this is to administer both systemic chemotherapy and neoadjuvant chemoradiotherapy prior to surgery to eliminate occult micrometastasis and increase the pCR rate [2]. The growing interest in implementing TNT for patients with LARC is fueled by the outcomes of published randomized controlled trials (RCTs) exploring TNT as a novel gold-standard approach to neoadjuvant chemoradiotherapy. These trials revealed a twofold increase in the pCR rate, a substantial decrease in the risk of subsequent metastases, and improved disease-free survival (DFS) and chemotherapy completion rates. Achieving pCR is crucial as it can offer patients the alternative of non-operative management, and in patients who undergo surgery, TNT could increase the likelihood of achieving rectal preservation [2]. Furthermore, TNT demonstrated the potential to abbreviate the conventional preoperative and postoperative adjuvant chemotherapy regimen from 15–19 weeks to 5 weeks preoperatively and from 6 months to 3 months postoperatively, thereby enhancing treatment adherence [3].

Several theories have been postulated to explain why TNT is superior to chemoradiation therapy in terms of improving outcomes in LARC patients. Firstly, TNT can achieve better local tumour control and hence eliminate micrometastatic disease. Secondly, TNT shortens the treatment duration and enhances adherence, potentially mitigating delays in surgery, improving treatment adherence and reducing operative complications and the rate of micrometastasis [1,4,5].

There are numerous potential TNT chemotherapy schedules, with no established gold-standard schedule available as of yet. These schedules encompass preoperative induction or consolidation chemotherapy with a variety of options, such as FOLFOXIRI, FOLFOX, or CAPEOX, with durations ranging from 6 to 18 weeks. Induction chemotherapy occurs prior to long-course chemoradiation (LCCRT), while consolidation neoadjuvant chemotherapy (NACT) occurs subsequent to either LCCRT or short-course preoperative radiation therapy (SCPRT). While still awaiting results from studies exploring the optimal TNT schedule, TNT offers promising outcomes for patients with LARC. Further data from RCTs on the effect of TNT on overall survival and quality of life are still pending.

The objective of this real-world retrospective analysis is to evaluate patient-centred outcomes, focusing on both the effectiveness and safety of TNT in patients diagnosed with LARC. This evaluation will be conducted within the context of two hospitals located in Western Australia.

## 2. Methods

### 2.1. Patient Population

This retrospective observational study included patients with newly diagnosed locally advanced rectal cancer treated with TNT in two centres in Western Australia between January 2018 and March 2023. The staging of the rectal cancer was discussed in a multidisciplinary team (MDT) meeting on the basis of histopathological analysis from biopsy samples obtained during colonoscopy and imaging, which included MRI of the rectum and an FDG-PET scan. Patients with biopsy-proven rectal adenocarcinoma situated up to 15 cm from the anal verge were included. This study was approved by the St John of God Human Ethics Research Committee (Approval Code 1936 on 7 July 2022).

### 2.2. Treatment Regimen

All protocol-mandated preoperative treatment was delivered at Fiona Stanley Hospital and St John of God Midland Hospital, Perth, Western Australia. With regard to the TNT sequence regimen, patients received either induction chemotherapy with the FOLFOX regimen followed by long-course chemoradiotherapy or long-course chemoradiotherapy followed by consolidation chemotherapy with the CAPOX regimen. For the mFOLFOX6 regimen, 5-fluorouracil (400 mg/m^2^ IV bolus on day 1, then 1200 mg/m^2^/day for 2 days), oxaliplatin (85 mg/m^2^ IV over 2 h on day 1), and leucovorin (400 mg/m^2^ IV over 2 h on day 1) were administered every 2 weeks for 9 to 12 cycles, while for the CAPOX regimen, capecitabine (1000 mg/m^2^/12 h per os on days 1–14) and oxaliplatin (oxaliplatin 130 mg/m^2^ IV over 2 h on day 1) were administered every three weeks for 6 cycles. Radiotherapy was scheduled 2 weeks after the completion of induction chemotherapy. Then, 45 Gy was given to the pelvis in 25 fractions, followed by the boost to the tumour to a dose of 50.4 Gy for T3 and 54 Gy for T4 tumours in 3–5 fractions. Concomitant chemotherapy with capecitabine was administered from the first to the last day of the radiation treatment (excluding weekends) at a daily dose of 825 mg/m^2^/12 h. Continuous 5-fluorouracil protracted infusion was given concurrently with radiotherapy at 225 mg/m^2^/day. Then, 8–10 weeks following the end of TNT, the progress of each patient was discussed in the colorectal multidisciplinary team meeting to decide if operative or non-operative management was appropriate. Operative management involved either abdominoperineal or sphincter-preserving surgery. No adjuvant therapy was administered post-surgery in patients who chose operative management.

### 2.3. Efficacy and Tolerability Endpoints

The baseline demographic characteristics of each patient included their age, sex, ECOG performance status, distance of the rectal mass from the anal verge, TNM rectal cancer stage, and TNT sequence regimen. The staging of rectal cancer was conducted in the MDT meeting using the American Joint Committee on Cancer (AJCC) 7th edition staging system.

The efficacy endpoints utilized for this study include radiological response rate (complete vs partial vs no response), pathological complete response rate, the proportion of patients who underwent surgery who had R0 resection vs R1 resection, the proportion of patients who underwent surgery who had abdominoperineal compared to anterior resection or sphincter-preserving surgery, the proportion of patients who had a radiological complete response and opted for a watch-and-wait approach instead of proceeding to surgery, and progression-free survival (PFS). We defined pCR as ypT0N0 (Dworak tumour regression grade 4)—indicating the absence of residual viable tumour cells in the surgical specimen.

To assess the safety and tolerability of TNT, the endpoints assessed included chemotherapy and radiotherapy completion rates, the proportion of patients with dose reduction and/or delay, and the proportion of patients who experienced common adverse effects due to chemotherapy and radiotherapy.

A subgroup analysis of the efficacy outcomes between the two treatment regimens (induction chemotherapy followed by chemoradiotherapy and chemoradiotherapy followed by consolidation chemotherapy)—specifically concerning the proportion of patients with complete clinical or partial clinical responses, the rate of R0 resection, and the pCR rate—was conducted.

## 3. Results

### 3.1. Baseline Demographics

Of the 32 patients with LARC treated with TNT, 20 (63%) of the patient population were male. Twelve (37%) were female. The median age was 60. In regard to ECOG performance status, 11 (35%), 19 (59%), and 2 (6%) patients were ECOG 0, 1, and 2, respectively (see Table 1). Seventeen (53%), ten (31%), and five (16%) of the patients had rectal tumours that were <5 cm, between 5 and 10 cm, and greater than 10 cm from the anal verge, respectively. One (3%), two (6%), twelve (38%), and seventeen (53%) of the patients had stage 2a, 3a, 3b, and 3c rectal tumours, respectively.

Seventeen patients (53%) received chemoradiotherapy followed by consolidation chemotherapy. Fifteen patients (47%) received induction chemotherapy followed by chemoradiotherapy.

Twelve patients (38%) received CAPOX as the chemotherapy regimen. Twenty patients (62%) received FOLFOX as the chemotherapy regimen.

### 3.2. Efficacy of TNT

The median duration of follow-up was 24.7 months. Nine (28%) of patients with LARC treated with TNT had a complete clinical response, of which five (56%) declined surgery and opted for a “watch and wait” approach (see Figure 1). Twenty-one (66%) patients had a partial clinical response, and two (6%) patients had no response to TNT. Of the 32 patients, 27 (84%) underwent surgery. All of the patients had R0 resection. The pCR rate was 15%. In regard to the type of surgery, 19 (70%) patients had sphincter-preserving ultralow anterior resection, whereas 8 patients (30%) had an abdominoperineal resection. The median PFS was not reached.

### 3.3. Comparison between the Efficacy of Chemoradiotherapy Followed by Consolidation Chemotherapy and the Efficacy of Induction Chemotherapy Followed by Chemoradiotherapy

Five (29%) of the patients with LARC treated with chemoradiotherapy followed by consolidation chemotherapy had a complete clinical response (see Table 2). Ten (59%) patients had a partial clinical response, and two (12%) patients had no response. The R0 resection rate was 100%. The pCR rate was 14%. Four (25%) of the patients with LARC treated with induction chemotherapy followed by chemoradiotherapy had a complete clinical response, and twelve (75%) patients had a partial clinical response. The R0 resection rate was 100%. The pCR rate was 15%.

### 3.4. Safety and Tolerability of TNT

In regard to the tolerability of TNT, 28 (88%) completed the chemotherapy course, although 19 (59%) required dose reduction and/or delay due to adverse effects relating to chemotherapy (see Table 3). In relation to adverse effects, most were grade 1 and 2. Overall, 30% of patients had grade 1 and 2 anemia, and 9% had grade ≥3 anemia; 21% of patients had grade 1 and 2 neutropenia, and 3% of patients had grade ≥3 neutropenia; 15% had grade 1 and 2 thrombocytopenia; 9% had grade 1 and 2 diarrhea; and 58% had grade 1 and 2 peripheral neuropathy. In total, 4 (12%) of the 33 patients had grade ≥3 toxicity, requiring treatment discontinuation.

## 4. Discussion

Although recent studies exploring the role of TNT in the treatment of LARC have yielded promising results, the optimal management of LARC remains unclear. Following the publication of results from the PRDIGE-23 and RAPIDO trials, in which TNT produced lower disease-related treatment failure, superior PFS, pCR rates, and chemotherapy completion rates, there is increased enthusiasm for delivering TNT for patients with LARC, especially those with high-risk features. Despite this, some questions remain regarding the optimal TNT sequence regimen, radiotherapy dose fraction method, and the optimal timing between chemotherapy and surgery. Additionally, there is some hesitation to adopt TNT as the standard of care for LARC due to concerns relating to increased toxicity.

In this study, we sought to confirm the efficacy and tolerability of TNT in LARC in our cohort of patients with LARC treated with TNT in two centres in Western Australia.

Our study highlights that TNT produced a substantial pCR rate of 15% when compared to the pCR rate of 20–40% achieved in published major trials, including RAPIDO, PRODIGE-23, CAO/ARO/AIO-12, and STELLAR trials [1]. Additionally, our study demonstrated that the R0 resection rate in our cohort of LARC patients was 100%, which is higher than the published R0 resection rates of 80–95% [3,6]. Achieving an optimal pCR and R0 resection rate is important as both are associated with improved PFS and OS. Furthermore, the efficacy of adjuvant chemotherapy following preoperative LCCRT/SCRT remains a subject of debate, with no demonstrated enhancement in overall survival (OS), which is partly attributable to suboptimal adherence to the treatment protocol [7,8,9]. This, along with the fact that 25% to 30% of advanced rectal cancer patients experience tumour-distant recurrence at 5 years despite adjuvant chemotherapy, makes the prospect of administering neoadjuvant chemotherapy a more appealing strategy [10].

Secondly, our study demonstrated that TNT produced a cCR of 28% in our cohort of LARC patients, similar to the cCR obtained from previous studies, ranging from 25 to 37% [11]. Attaining cCR will provide the alternative of using a watch-and-wait (WW) strategy instead of surgery, which is associated with significant morbidity and impacts patient quality of life [12]. Considering the significant improvements in survival outcomes in LARC patients treated with TNT, the WW strategy has gained acceptance as a potential treatment option in selected cases in recent years, helping patients avoid surgery. In our cohort, 50% of patients who achieved cCR opted for the WW approach. This is comparable to a study conducted by Gani et al. (2019) in which 83% of patients expressed a preference for a WW strategy in the event of achieving cCR, while acknowledging the risk of recurrence [13]. Interestingly, up to 15% of patients who had an incomplete clinical response (iCR) ended up having a pCR at the time of histopathologic analysis. In our cohort, only 1 of the 24 patients (4%) who had an iCR ended up having a pCR. Thus, establishing a precise definition of cCR is crucial in terms of promoting the acceptance of WW strategies while ensuring oncological safety. Currently, the WW strategy is not considered to be the standard of care. However, an increasing body of evidence may advocate for its implementation in specialized high-volume centres for specific patient cohorts.

Thirdly, we confirmed the efficacy of TNT in downstaging and downsizing tumours in LARC patients. In total, 70% of the patients who underwent operative management had lower anterior resection (LAR), most of which had the loop ileostomy reversed. Furthermore, 30% of LARC patients in our cohort had to undergo an APR. Our proportion of patients who had LAR falls short of the 82% rate seen in a retrospective study conducted by Greco et al. (2023). However, it is noteworthy that their cohort of patients received consolidation neoadjuvant chemotherapy in addition to induction chemotherapy [14]. In patients where operative management is needed, such as patients with incomplete clinical response and high-risk disease (i.e., extramural venous invasion and extensive mesorectal/pelvic lymph nodes involvement), sphincter-sparing surgery should be offered where feasible to optimize their quality of life. TNT allows patients who were previously planned to undergo abdominal perineal resection (APR) due to the proximity of the tumour to the anal sphincter to be offered a LAR, sparing patients from a permanent colostomy. Previous studies have confirmed that the presence of stoma negatively impacts the quality of life, thereby highlighting the important role of TNT in downsizing tumours to facilitate organ preservation [15].

Our study also demonstrated that there was no significant difference in the efficacy outcomes between the two TNT regimens. Specifically, the induction and consolidation chemotherapy regimen produced a similar clinical response, R0 resection rate, and pCR in patients with LARC. At present, there is a lack of consensus regarding the optimum sequence of TNT. However, a recent meta-analysis of 27 randomized clinical trials (a total of 13,413 individuals) conducted by Turri et al. (2024) demonstrated that long-course chemoradiotherapy followed by consolidation chemotherapy provides the greatest benefit in regard to achieving a pCR compared to the other two TNT regimens, which include short-course chemoradiotherapy followed by consolidation chemotherapy and induction chemotherapy followed by long-course chemoradiotherapy [16]. Furthermore, the optimal duration of treatment remains to be determined. This is critical as it determines adherence to treatment. Our study showed that chemotherapy completion rates were similar between the two groups (81% induction vs. 88% consolidation), concurring with results obtained from the OPRA trial (82% induction vs. 81% consolidation) [17]. At present, identifying patients who are most likely to benefit from a specific TNT regimen poses a challenge due to the lack of robust data on this matter.

In regard to tolerability and safety, we found that TNT is a safe and tolerable treatment strategy for patients with LARC. The chemotherapy completion rate of 88% in our study concurs with the 80–90% rate obtained in prior TNT studies focusing on LARC patients [3,6]. Although most patients completed the chemotherapy portion, consistent with previous studies, a significant proportion required dose reduction and/or treatment delay. In fact, the proportion of patients requiring dose reduction/delay in our cohort was lower at 59% compared to the 86% rate observed in the STELLAR trial [18]. The most common adverse events observed with TNT included peripheral neuropathy, gastrointestinal issues (such as diarrhea and nausea), hematologic complications (such as anemia, thrombocytopenia, and neutropenia), and fatigue. Most of the observed adverse effects were grade 1 and 2, which is consistent with previous trials [3,6,18]. The spectrum of toxicities was as expected in relation to the therapy administered. While grade 3 and 4 toxicities in patients treated with TNT are generally higher compared to standard chemoradiotherapy, they are still lower than toxicities observed in patients treated with adjuvant chemotherapy.

The limitations of our study include the small sample size derived from only two centres, the lack of a standardized control arm to perform statistical analysis to compare the efficacy and safety of TNT, and the short follow-up of our cohort of LARC patients receiving TNT. Additionally, we did not conduct a subgroup analysis of the efficacy and tolerability of various TNT regimens. Furthermore, no correlation between clinicopathologic features and other biomarkers, such as mismatch repair deficiency status, was established with respect to efficacy and tolerability. This would be valuable in terms of identifying the subset of patients who would benefit the most from TNT.

Future directions in the area could include conducting studies analyzing the optimum combination, sequence, or duration of TNT that would deliver the highest efficacy while minimizing toxicity; developing various biomarkers that would allow clinicians to individuate patients benefiting most from TNT; and studying novel approaches, such as combining chemotherapy with immunotherapy, to further improve outcomes in LARC patients, especially those who are poor responders to TNT.

## 5. Conclusions

In conclusion, TNT has emerged as a pivotal strategy in the comprehensive management of LARC. The principal merit of TNT lies in its profound impact on achieving cCR and augmenting pCR. These favourable outcomes translate into a notable improvement in disease-free survival and likely extend to overall survival, accompanied by a significantly enhanced quality of life through the avoidance of a permanent colostomy. Despite its potential, the optimal TNT regimen remains to be precisely defined, necessitating a nuanced, stratified approach. Ongoing advancements across all facets of treatment modalities involved in TNT hold considerable promise for further elevating the therapeutic outcomes for individuals afflicted by rectal cancer.

## Figures and Tables

**Figure 1 diseases-12-00257-f001:**
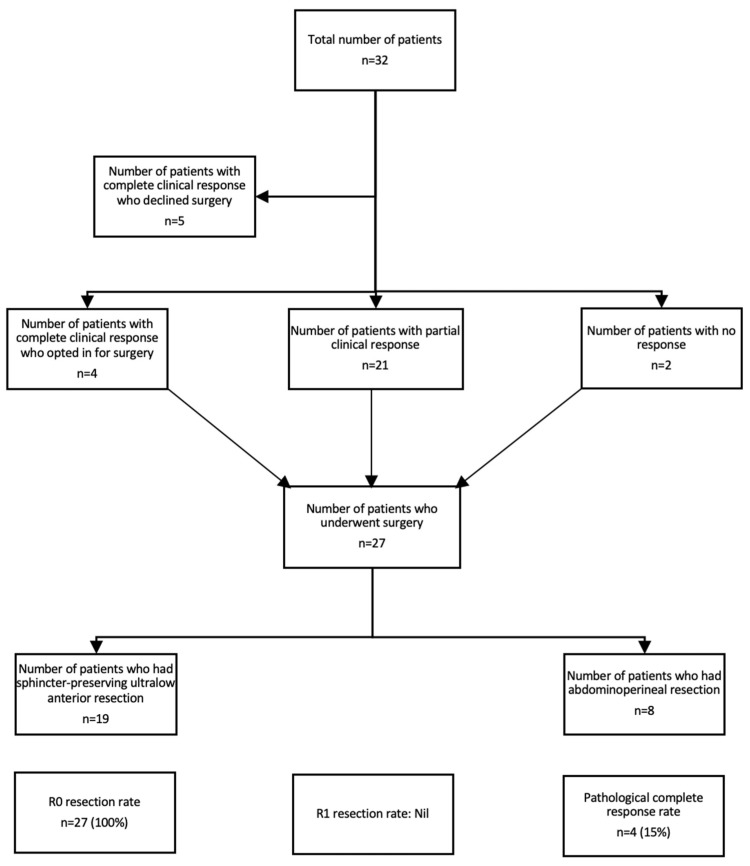
CONSORT diagram demonstrating the efficacy of TNT.

**Table 1 diseases-12-00257-t001:** Baseline demographics of the selected patients.

Baseline Demographics	*n* (%)
Total number of patients	32
Male	20 (63%)
Female	12 (37%)
Median age	60
ECOG performance status	
ECOG 0	11 (35%)
ECOG 1	19 (59%)
ECOG 2	2 (6%)
Rectal tumour distance from anal verge	
<5 cm	17 (53%)
5–10 cm	10 (31%)
>10 cm	5 (16%)
Rectal tumour stage	
2a	1 (3%)
3a	2 (6%)
3b	12 (38%)
3c	17 (53%)
TNT sequence	
Chemoradiotherapy followed by consolidation chemotherapy	17 (53%)
Induction chemotherapy followed by chemotherapy	15 (47%)
Chemotherapy regimen	
CAPOX	12 (38%)
FOLFOX	20 (62%)

**Table 2 diseases-12-00257-t002:** Comparison between the efficacy of TNT with consolidation chemotherapy and with induction chemotherapy.

Efficacy of TNT	With Consolidation Chemotherapy *n* (%)	With Induction Chemotherapy *n* (%)
Clinical complete response	5 (29%)	4 (25%)
Clinical partial response	10 (59%)	12 (75%)
No clinical response	2 (12%)	0 (0%)
R0 resection rate	14 (100%)	13 (100%)
Pathological complete response rate	2 (14%)	2 (15%)
Chemotherapy course completion rate	15(88%)	13 (81%)
Radiotherapy course completion rate	16 (94%)	15 (94%)

**Table 3 diseases-12-00257-t003:** Safety and tolerability of TNT.

Safety and Tolerability of TNT		*n* (%)	
Patients who completed the chemotherapy course		28 (88%)	
Patients who required dose reduction and/or delay due to adverse effects		19 (59%)	
Patients who required treatment discontinuation due to severe adverse events		4 (12%)	
Patients who completed radiotherapy course		28 (85%)	
Adverse effects	Grade 1 and 2		Grade ≥ 3
Anaemia	10 (30%)		3 (9%)
Neutropenia	7 (21%)		1 (3%)
Thrombocytopenia	5 (15%)		
Diarrhea	3 (9%)		
Peripheral neuropathy	19 (58%)		

## Data Availability

The original contributions presented in the study are included in the article. Further inquiries can be directed to the corresponding author/s.

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
