# Peer review of "Total Neoadjuvant Therapy in Locally Advanced Rectal Cancer: Insights from the Western Australian Context"

_diseases, 2024, doi:10.3390/diseases12100257_

Round 1
Reviewer 1 Report (Previous Reviewer 2)
Comments and Suggestions for Authors
Anything more to say about the paper . I think it has rigurosity .
Author Response
Thank you for the suggestion.
Reviewer 2 Report (Previous Reviewer 1)
Comments and Suggestions for Authors
The authors have responded only to the minor suggestions related to some technical aspects of the manuscript. The major concern remains: although the topic is potentially interesting as it refers to a relatively novel therapeutic approach to locally advanced rectal cancer, the significance of the results is questionable considering the sample size and the study fails to produce sufficient originality to contribute to the knowledge that could support wider use of this treatment regimen. Also, the patient recruitment strategy remains unexplained, as well as the strategy for the decision-making on operative or non-operative management. The text is still doubling a lot of data from the tables.
Comments on the Quality of English LanguageNo major issues detected
Author Response
We recognize that the limited sample size in our study constitutes a significant limitation. In future research, we plan to increase patient recruitment to validate our findings. Expanding the sample size will also facilitate subgroup analyses of various total neoadjuvant therapy regimens for the treatment of locally advanced rectal cancer (LARC) patients. With total neoadjuvant therapy now established as the gold standard for LARC, we anticipate it will be more feasible to recruit a larger cohort moving forward.
As previously addressed, the decision regarding whether LARC patients who had complete clinical response should proceed with surgery or active surveillance is determined by discussions in the multidisciplinary team meeting, where prognostic factors such as patient comorbidities, pretreatment staging, and lymph node involvement are carefully evaluated. Following this, the final decision regarding which of the two approaches is utilised is determined through a shared decision-making process between the clinician and the patient.
We have modified our table to ensure that repetition of data is minimised.
Reviewer 3 Report (New Reviewer)
Comments and Suggestions for Authors
The authors present their findings of real-life Western Australian data well with limitations clearly noted. I would recommend the authors reference this recent paper which outlines that TNT regimens should be recognised as first-line treatments when aiming at increasing pCR rates in LARC. Specifically there are 3 TNT protocols that have been shown to outperform standard of care and the manuscript should be revised accordingly to reflect this. https://jamanetwork.com/journals/jamanetworkopen/fullarticle/2819451
Author Response
Thank you for the suggestion. Remarks highlighting the latest data on the most efficacious TNT regimen as suggested by reviewer 3 has been added to the manuscript. Please see line 245 to 250.
Round 2
Reviewer 2 Report (Previous Reviewer 1)
Comments and Suggestions for Authors
The resubmitted manuscript shows only minor changes from the initial submission after two review rounds, with the authors making only slight revisions to the text. The primary concern remains the questionable significance of the results given the sample size. Additionally, the study lacks sufficient originality to contribute meaningfully to the knowledge needed to support broader use of the specific treatment regimen investigated. While the authors acknowledge the study's limitations and plan to address them in future research by recruiting a larger patient cohort and conducting a more comprehensive analysis of regimen subtypes, this does not adequately address the raised concerns. As it stands, the manuscript reflects a single-center experience with a novel therapeutic regimen and is better suited for a local or regional journal.
Comments on the Quality of English LanguageNo major issues detected
Author Response
We recognize that the limited sample size in our study constitutes a significant limitation. In future research, we plan to increase patient recruitment to validate our findings. Expanding the sample size will also facilitate subgroup analyses of various total neoadjuvant therapy regimens for the treatment of locally advanced rectal cancer (LARC) patients. With total neoadjuvant therapy now established as the gold standard for LARC, we anticipate it will be more feasible to recruit a larger cohort moving forward.
As previously addressed, the decision regarding whether LARC patients who had complete clinical response should proceed with surgery or active surveillance is determined by discussions in the multidisciplinary team meeting, where prognostic factors such as patient comorbidities, pretreatment staging, and lymph node involvement are carefully evaluated. Following this, the final decision regarding which of the two approaches is utilised is determined through a shared decision-making process between the clinician and the patient.
We have modified our table to ensure that repetition of data is minimised.
This manuscript is a resubmission of an earlier submission. The following is a list of the peer review reports and author responses from that submission.
Round 1
Reviewer 1 Report
Comments and Suggestions for Authors
The manuscript focuses on relatively novel therapeutic approach to locally advanced rectal cancer, but presented results lack the necessary statistical power and the study fails to contribute significantly to the knowledge that could support wider use of this treatment regimen.
The major issues are the study size and design. The patient group is small and insufficient to produce strong conclusions on efficacy and safety of the treatment outcomes. Also, subgroup analysis of the efficacy and tolerability of various regimen subtypes could not be conducted.
Additionally, the patient recruitment strategy is not clearly presented. It is not clear if the patients were recruited to opt for the analyzed treatment or the treatment was applied as a part of routine clinical management. The strategy for the decision making on operative or non-operative management is also questionable, as the manuscript states that it was both decision of the multidisciplinary team and the choice of patients. It remains unclear if there were any specific criteria involved in this decision.
The manuscript also lacks technical clarity. The abstract is too long and not very informative. The text of the Results section is burdened with numbers that double the information from the tables. Comparison of consolidaiton vs. induction therapy should have been given in the same table, not separate ones. Also, the numbers and the percentages in some of the tables are confusing due to the fact that five patients who opted for watch and wait are excluded from some summations.
Comments on the Quality of English LanguageNo major issues detected
Author Response
The patient group is small and insufficient to produce strong conclusions on efficacy and safety of the treatment outcomes. Also, subgroup analysis of the efficacy and tolerability of various regimen subtypes could not be conducted.
Response: Thank you for the comment. We agree with this comment – our study is a case series of only 32 patients and as such, it is difficult to draw strong conclusions on the efficacy and safety of total neoadjuvant therapy (TNT) and certainly, impossible to conduct formal statistical subgroup analysis on various regimen subtypes.
The patient recruitment strategy is not clearly presented. It is not clear if the patients were recruited to opt for the analyzed treatment or the treatment was applied as a part of routine clinical management.
Response: The patients treated at our cancer centre at the time was offered the gold srandard treatment of preoperative concurrent chemoradiation or the novel TNT. Patients who opted for TNT were included in this case series.
The strategy for the decision making on operative or non-operative management is also questionable, as the manuscript states that it was both decision of the multidisciplinary team and the choice of patients. It remains unclear if there were any specific criteria involved in this decision.
Response: The decision as to whether an operative or non-operative management approach was adopted was dependent on numerous factors. These include clinicopathological factors such as patients’ response to TNT, pre-operative cancer stage, tumour mutation status, patients’ age and comorbidities. These factors are discussed in a multidiscplinary team meeting involving surgeons, medical oncologists, radiation oncologists, radiologists and pathologists. Provided that there are no contraindications, patients with a complete or near-complete clinical response to TNT, are offered a watch and wait strategy given patients with a complete or near-complete clinical response to TNT have comparable overall and disease-free survival to their counterparts who undergo surgical resection, and also have a better quality of life, fewer complications, and potentially avoid a stoma.
The abstract is too long and not very informative. The text of the Results section is burdened with numbers that double the information from the tables.
Response: Thank you. Please see revised abstract – this has been addressed.
Comparison of consolidation vs. induction therapy should have been given in the same table, not separate ones. Also, the numbers and the percentages in some of the tables are confusing due to the fact that five patients who opted for watch and wait are excluded from some summations.
Response: Thank you. Please see the revised comparison table in the updated manuscript. We have changed one of the tables into a consort diagram to demonstrate our data more concisely and clearly.
Reviewer 2 Report
Comments and Suggestions for Authors
Dear author,
congratulations on the work done.
The implementation of the TNT is a very current issue and it is always good to have rigorous information on the results obtained by different groups.
I would like to expand on different issues:
- the study population is 32 patients, but I would like to know if there were any patients who could be candidates and who were rejected and why.
-- How did the patients who underwent watch and wait evolve?
- Does the full Pathological Response match the full Clinical?
- It talks with the adverse effects of TNT but what about the adverse effects of surgery?
Many thanks and congratulations again
Author Response
the study population is 32 patients, but I would like to know if there were any patients who could be candidates and who were rejected and why.
Response: The promising efficacy and safety of TNT has became more established in the past one to two years. At the time of study commencement, the respective cancer centre from which the patients were from utilised preoperative concurrent chemoradiation as the gold standard treatment for locally advanced rectal cancer patients, with TNT only emerging as a promising, novel treatment. As such, we had only few patients who were treated with TNT.
How did the patients who underwent watch and wait evolve?
Response: The 5 patients who underwent a watch and wait strategy remained in remission at the time of their last follow up with the last date of data collection being 31 December 2023.
- Does the full Pathological Response match the full Clinical?
Response: Four of the nine patients who had a clinical complete response underwent surgery. Two of the four patients who had a clinical complete response did not have a pathological complete response response, the other two patients who had a clinical complete response also had a pathological complete response.
- It talks with the adverse effects of TNT but what about the adverse effects of surgery?
Response: Unfortunately, in this study we did not collect data on the adverse effects of surgery. We recognise that this is another limitation of our study.
Round 2
Reviewer 1 Report
Comments and Suggestions for Authors
The authors haven’t really included any of the relevant responses in the manuscript. The major issues for concern remain – the size of the group is insufficient for strong conclusions, and the recruitment strategy and clinical management of the patients also remain unclear.
Author Response
Thank you for your feedback. We acknowledge the concern regarding the size and design of our study. We recognize that the small number of patients limits our ability to draw robust conclusions about the efficacy and safety of the treatment, and that a subgroup analysis of regimen subtypes was not feasible within the scope of this study. We are aware of these limitations and plan to address them in future research by recruiting a larger patient cohort and incorporating a more comprehensive analysis of regimen subtypes. We have addressed the remaining concerns in our response letter, and we have made the corresponding changes regarding the length of the abstract and data table.